# From Chemistry to Bioactivity: HS-SPME-GC-MS Profiling and Bacterial Growth Inhibition of Three Different Propolis Samples from Romania, Australia, and Uruguay

**DOI:** 10.3390/molecules30194014

**Published:** 2025-10-08

**Authors:** Radosław Balwierz, Katarzyna Kasperkiewicz, Martyna Straszak, Daria Siodłak, Katarzyna Pokajewicz, Ibtissem Ben Hammouda, Piotr P. Wieczorek, Anna Kurek-Górecka, Zenon P. Czuba, Tomasz Baj

**Affiliations:** 1Institute of Chemistry, University of Opole, Oleska 48, 45-052 Opole, Poland; 2Faculty of Natural Sciences, Institute of Biology, Biotechnology and Environmental Protection, University of Silesia in Katowice, Jagiellonska 28, 40-032 Katowice, Poland; 3Department of Microbiology and Immunology, Faculty of Medical Sciences in Zabrze, Medical University of Silesia in Katowice, Jordana 19, 41-808 Zabrze, Poland; 4Department of Pharmacognosy with the Medicinal Plant Garden, Medical University of Lublin, Chodzki 1, 20-093 Lublin, Poland

**Keywords:** propolis, antibacterial activity, volatile organic compounds, geographic origin, *Staphylococcus aureus*, *Streptococcus mutans*

## Abstract

Propolis is a valuable natural product whose chemical composition and bioactivity are strongly dependent on its geographical and botanical origin. This study presents a comprehensive comparative analysis of the volatile profiles and antibacterial properties of propolis from Romania, Australia, and Uruguay, benchmarked against previously published data from samples from Poland and Turkey. Volatile compounds were profiled using headspace solid-phase microextraction coupled with gas chromatography-mass spectrometry. The resulting data were interrogated using multivariate chemometric analyses (HCA, PCA), and antibacterial activity was assessed via the disk diffusion method against five bacterial strains. Chemometric analysis revealed a clear demarcation into two primary chemotypes: a European type (Poland, Romania, Turkey) dominated by aromatic compounds such as benzoic acid, and a non-European type (Australia, Uruguay) characterized by a high abundance of terpenes. The Australian propolis exhibited a complex terpene profile rich in α-copaene and pinenes, while the Uruguayan sample was distinguished by an exceptionally high concentration of α-pinene. All active extracts showed selective, concentration-dependent inhibition against Gram-positive *Staphylococcus aureus* and *Streptococcus mutans*. The terpene-rich Australian propolis displayed the highest antibacterial potency, particularly against *S. mutans*. Crucially, Pearson correlation analysis revealed a counter-intuitive relationship: the most abundant terpenes in the non-European samples (e.g., α-pinene, verbenone) were significantly negatively correlated with antibacterial activity (r ≈ −0.99). Conversely, less abundant compounds, including linalool and acetic acid, were identified as strong positive predictors of inhibition (r > 0.90). These findings underscore a complex geography-chemotype-bioactivity relationship, where the overall synergistic effect of a mixed chemical profile, rather than the dominance of a single compound, determines antibacterial potency. The initially proposed markers provide a basis for origin-based standardization and highlight Australian propolis as a promising source of natural antibacterial agents.

## 1. Introduction

Propolis, commonly referred to as “bee glue”, is a natural resinous substance produced by honeybees (*Apis mellifera*) through the collection of plant exudates from buds, bark, and leaves. These materials are subsequently combined with wax and salivary enzymes that provide modifying and protective functions of bees. Propolis serves as both a construction and protective material, ensuring aseptic conditions in the hive by sealing gaps and limiting microbial growth [1]. Its chemical composition is remarkably complex and environmentally variable, encompassing more than 800 identified compounds, including flavonoids (over 140 compounds), phenolic acids and their esters, terpenes and terpenoids (over 160 compounds), fatty acids, waxes, essential oils, sugars, vitamins, and trace elements [1,2]. This chemical diversity translates into a broad spectrum of biological activities, such as antimicrobial, anti-inflammatory, antioxidant, anticancer, immunomodulatory, and potentially antidiabetic effects [2,3]. Consequently, propolis has gained significant pharmaceutical and nutraceutical relevance. It is currently incorporated into a wide range of commercial preparations worldwide, including ethanolic extracts, capsules, tablets, and complex formulations [3,4].

The chemical composition and bioactivity of propolis are strictly dependent on its geographical and botanical origin [1]. Several types of propolis can be distinguished according to their geographical source and chemical profile: temperate (e.g., Europe, North America, Western Asia—rich in flavonoids and phenolic acid esters), poplar (e.g., Poland, Germany, USA—containing flavones, flavanones, and phenolic acids), birch (Russia, Poland—characterized by specific flavones and flavonols), Mediterranean (e.g., Greece, Croatia, Algeria—high diterpenoid and low flavonoid content), Pacific (Taiwan, Japan—dominated by prenylated flavanones), tropical (e.g., Brazil, Venezuela—containing prenylated *p*-coumaric acid derivatives, benzophenones, lignans, and terpenes), Brazilian [green from *Baccharis dracunculifolia* (phenolic acids, triterpenoids), red from *Dalbergia ecastaphyllum* and *Clusia* spp. (isoflavonoids, benzophenones), and brown], Cuban (brown, yellow, red), and Indian (propolis from northern and southern regions of the country) [5]. For example, poplar-type propolis, characteristic of the temperate zones of Europe, including Poland and Romania, is rich in flavonoids and phenolic acids, which determine its strong antioxidant and antibacterial activity [5,6]. In Poland, beyond black poplar, honey bees also obtain propolis resins from aspen (*Populus tremula*) and from the leaf-bud secretions of birches (*Betula* spp.) and alders (*Alnus* spp.); the latter two are confirmed for Polish propolis by microscopic identification of secretory discs matching *Betula*/*Alnus* leaves, and the aspen/birch contribution is supported by chemical-botanical correlation in Polish datasets [7,8]. Romanian propolis is also classified as a poplar-type propolis. Its principal botanical source is resin from *Populus* species; however, depending on the collection region, resins from *Quercus*, *Aesculus*, *Ulmus*, *Picea*, *Salix*, and *Fraxinus* may also contribute. Across studies, Romanian propolis extracts typically contain an average of 250–300 mg/g of polyphenolic compounds, with substantial geographic variability [5]. Turkish propolis is distinguished by its high content of caffeic acid phenethyl ester (CAPE) and chrysin, which translates into strong antioxidant and anti-inflammatory activity [9,10]. Typical characteristic flora used by bees to produce Turkish propolis includes: *Populus* spp., *Pinus* spp., *Castanea* spp., *Quercus* spp., *Tilia* spp., *Helianthus annuus* L., *Salix* spp., *Juniperus* spp., *Acer* spp., *Rhododendron* spp., *Citrus* spp., *Betula* spp., *Acacia* spp. [9]. In contrast, Australian propolis, produced with the involvement of stingless bees and eucalyptus plants, contains unique prenylated flavonoids that demonstrate antidiabetic and neuroprotective properties [11,12]. In Australia, the main source of resin for bees is *Eucalyptus*, *Corymbia* and regionally *Angophora*. This reflects the ecological dominance of *Myrtaceae* in Australian floras and landscapes [13]. *Corymbia torelliana*, known as cadaghi fruit resin, is a documented source for stingless-bee propolis (*Tetragonula carbonaria*) [12]. In southern Australia, a “single-plant-source” bee glue type has been linked to *Myoporum insulare* (*Scrophulariaceae*) [14]. Uruguayan propolis, although typologically close to the European type, is characterized by a relatively low phenolic content, which accounts for its weaker antioxidant activity and makes it the least studied among the mentioned types. The Uruguayan native flora includes approximately 2500 species of plants, and it is divided into five groups: Monte Ribereño, Monte de Parque, Monte de Quebrada, Monte Serrano and Planted Forest as non-native flora, including species like *Eucalyptus grandis*, *Eucalyptus Globulus* ssp. *globulus*, *Eucalyptus Globulus* ssp. *Maidenii*, *Eucalyptus Dunnii*, *Pinus Elliotte* and *Pinus Taeda* [5,15].

Despite extensive research, significant knowledge gaps remain regarding propolis from certain regions. In the case of Romanian propolis, studies explaining the molecular mechanism of action are needed, although propolis is being used in complementary and alternative medicine as support of classical medicine [16,17]. In Australia, only fragmentary data are available, primarily addressing the mechanism of action in vitro and in silico studies, without confirmation in in vivo experiments [18,19]. Uruguay, the major issue is the near absence of data concerning botanical sources, biological activity, and potential pharmacological applications [20,21]. Moreover, comparative studies between regions suffer from a lack of methodological standardization—different extraction protocols, analytical techniques (GC-MS, HPLC, LC-MS), and bioactivity models are applied—making it difficult to integrate findings consistently [22,23,24].

Importantly, gas chromatography coupled with mass spectrometry (GC-MS) remains the primary chemotaxonomic tool in propolis research, enabling the identification of botanical and geographical markers as well as the creation of characteristic metabolic profiles [22,25]. GC-MS allows the detection of volatile compounds such as α-pinene, β-pinene, limonene, α-copaene, and methyl salicylate, which may correlate with antimicrobial activity [26]. Multivariate analysis of GC-MS data, for example, principal component analysis (PCA), enables the differentiation of samples from various regions and the assessment of chemotype similarities [27,28]. However, the limitation of this technique is that it captures only the volatile and semi-volatile fraction of compounds, meaning that the full composition of ethanolic extracts—commonly used in biological studies—is not fully represented [27,28].

The aim of this study is to present preliminary comparative investigations of the volatile composition and antibacterial activity of three propolis samples originating from Romania, Australia, and Uruguay, analyzed in relation to previously published data from Poland and Turkey [9], as well as standardized extract of propolis from Uruguay. The research involved GC-MS analysis of raw propolis samples, followed by disk diffusion assays of ethanolic extracts against selected reference bacterial strains. Due to different analytical procedures and the limited amount of material available, solid phase microextraction from headspace followed by gas chromatography-mass spectrometry (HS-SPME-GC-MS) analysis was performed on raw samples, whereas biological activity was assessed using extracts; therefore, the correlations between chemical composition and activity are qualitative in nature. These investigations thus serve as a starting point for future in-depth analyses, including quantitative determinations. The working hypothesis is that the presence of specific volatile compounds (monoterpenes and sesquiterpenes) together with phenolic acids may explain the observed antimicrobial activity, particularly against Gram-positive bacteria [6,29].

## 2. Results

### 2.1. HS-SPME-GC-MS Analysis of Volatile Compounds in Propolis

In the whole project, a total of 175 volatile compounds were identified and relatively quantified in volatile profiles of five propolis samples collected from different geographical regions: Poland (sample 01), Turkey (sample 02), Romania (sample 03), Australia (sample 04), and Uruguay (sample 06). The results of the study of Polish and Turkish propolis (samples 01–02) have already been published [9]. The HS-SPME-GC-MS analysis revealed substantial chemodiversity, highlighting the critical influence of geographical origin and local flora on the volatile composition of propolis. Detailed results, including the relative percentage abundance of individual compounds, are presented in Table 1. A comparative characterization of the volatile profiles of propolis from Romania, Australia, and Uruguay was carried out for the first time, in relation to well-documented Polish and Turkish samples [9].

The results of the Hierarchical Cluster Analysis (HCA), presented as a dendrogram and agglomeration schedule (Figure 1), clearly demonstrate a strong clustering of propolis samples according to their geographical origin. Two major, distinctly separated clusters were identified. The analysis identified a broad European cluster, within which Polish and Romanian samples formed a very tight subcluster, reflecting their strong chemical similarity (linkage distance: 100.125). The more distinct Turkish sample subsequently joined this subcluster at a much greater distance (584.419), forming the final Eurasian grouping and highlighting its different chemotype. The non-European cluster consisted of the Australian and Uruguayan samples, which clustered together at a linkage distance of 495.405. The clear separation between these two principal groups was further confirmed by the final merging step at the highest linkage distance (2608.601), highlighting the fundamental differences in the chemical profiles of European propolis compared with samples from other continents.

### 2.2. Volatile Pprofile of Eurasian Samples (Romania, Poland, Turkey)

The volatile profile of Romanian propolis (sample 03) was dominated by aromatic compounds, resembling the reference sample from Poland [9]. The major constituent in both cases was benzoic acid, though in the Romanian sample its proportion was markedly higher, accounting for 31.43% of the total volatile fraction. Other relevant constituents in sample 3 included benzyl alcohol (5.76%), vanillin (3.00%), and limonene (1.98%). Although both Polish and Romanian propolis shared an aromatic-based profile, they differed markedly in their relative proportions. The Romanian sample contained much higher levels of vanillin and benzoic acid, while the Polish sample [9] was characterized by benzyl alcohol oxidation products such as benzaldehyde. In contrast, the cited above Turkish reference sample (sample 02) displayed a more balanced profile: alongside aromatic compounds (benzoic acid 10.55%, benzyl alcohol 4.17%), monoterpenes such as limonene (5.32%) [9] played an important role, clearly distinguishing it from the other European samples.

### 2.3. Unique Profiles of Propolis from Australia and Uruguay

The volatile fraction of Australian propolis was dominated by terpenes, primarily sesquiterpenes and monoterpenes. The most abundant compounds were α-copaene (12.92%), α-pinene (12.09%), and β-pinene (7.15%). Other relevant terpenoids included substance tentatively identified as verbenone (5.78%) and *cis*-calamenene (4.11%). Aromatic compounds were present only at lower levels: benzoic acid and traces of benzyl alcohol and phenyl ethyl alcohol. This terpene-rich composition stands in clear contrast to the European samples, in which benzoic acid dominated the volatile profile. The Uruguayan raw propolis showed a distinct chemical pattern, with α-pinene as the dominant constituent (28.89%). Other major monoterpenes included β-pinene (4.54%) and *p*-cymene (1.28%), along with sesquiterpenes such as *cis*-calamenene (2.01%) and α-copaene (2.51%). Aromatic compounds occurred only in small amounts, with benzyl alcohol (0.30%) and phenyl ethyl alcohol (0.13%) far lower than in European samples. This profile highlights α-pinene as a defining chemical marker for Uruguayan propolis and underscores its clear divergence from both European and Australian profiles. Both non-European samples exhibited terpene-dominated profiles but with different signatures: Australian propolis combined α-copaene with balanced levels of monoterpenes and other sesquiterpenes, while Uruguayan propolis was characterized by a strong dominance of α-pinene and supporting terpenes, making it chemically distinct.

To further illustrate the chemical variability and to highlight the regional markers differentiating the samples, the distribution of volatile compounds across all studied propolis specimens was visualized using a heatmap with hierarchical clustering (Figure 2).

Hierarchical cluster analysis presented as a heatmap (Figure 2) clearly indicates a distinct grouping of propolis samples according to their geographical origin. The European samples (Poland, Romania, Turkey) formed a compact cluster characterized by a high contribution of aromatic compounds. In contrast, the non-European samples (Australia and Uruguay) formed a separate branch of the dendrogram, distinct from the European cluster, highlighting their unique terpene-rich chemical profiles. The pronounced differences in colour intensity reflect marker compounds that determine the regional specificity of the individual samples.

### 2.4. Comparative Overview and Candidate Geographic Volatile Markers

Altogether, five volatile profiles were compared: three obtained experimentally in the present study and two extracted from previously published data [9]. Polish and Romanian samples formed an “aromatic group” typical of European poplar-type propolis. In contrast, Australian and Uruguayan samples formed a “terpene group,” though with different chemical compositions. The dominance of α-copaene and pinenes in the Australian sample, and α-pinene and β-pinene in the Uruguayan sample, supports their role as potential geographical markers. Several compounds were common across most samples but occurred in markedly different proportions. Benzaldehyde, limonene, α-pinene, β-pinene, and camphor were present in at least four of the six samples, suggesting their widespread occurrence in plant resins collected by bees worldwide. However, it is the relative abundance, not mere presence, that determines the uniqueness of a given propolis. For instance, α-pinene in the Polish sample accounted for only 0.44%, while in the Australian and Uruguayan samples, its proportion exceeded 12% and 28%, respectively. In conclusion, GC-MS analysis of volatile fractions demonstrated substantial chemodiversity strongly correlated with geographical origin. European propolis (Poland, Romania) exhibited aromatic-rich profiles, whereas Australian and Uruguayan samples were dominated by terpenes, each with region-specific signatures. It appears that dominant compounds, such as α-copaene in the Australian sample and α-pinene in the Uruguayan sample (with β-pinene also contributing as a relevant marker), could potentially serve as useful chemical indicators for authenticity and geographical origin verification. These distinctive volatile profiles are crucial for understanding the biological activity and therapeutic potential of propolis from diverse regions worldwide.

Principal Component Analysis (PCA) was applied to visualize relationships among the chemical profiles of the analyzed propolis samples and to identify volatile compounds responsible for the observed variability. The loadings scatterplot of the first two principal components (PC1 and PC2), which together explained 89.3% of the total variance, is shown in Figure 3.

PC1 (56.5% of variance) clearly separated the samples according to their dominant chemical type. European propolis (Poland, Romania, Turkey) clustered on the positive axis, strongly associated with aromatic compounds such as benzoic acid, benzyl alcohol, and vanillin, confirming their role as key markers of poplar-type propolis. Non-European samples (Australia, Uruguay) clustered on the negative axis, driven by high terpene content.

PC2 (32.8% of variance) further differentiated samples within these groups. Uruguay (high positive PC2 values) correlated strongly with α-pinene, establishing this monoterpene as its principal marker. Australia (negative PC2 values) was associated with sesquiterpene α-copaene and the monoterpenes β-pinene and limonene. The European cluster grouped Polish and Romanian samples. Within the European cluster, the PC2 axis clearly separated the Turkish sample from the tight Polish-Romanian group. This distinction was driven by Turkey’s relatively higher levels of menthol and acetic acid, underscoring its unique chemical profile among the Eurasian samples.

Overall, PCA fully confirmed the HCA results and enabled the identification of key chemical markers defining the geographical origin of propolis. Aromatic compounds determined the European profiles, whereas non-European samples were dominated by terpenes with region-specific signatures.

### 2.5. Antibacterial Activity

Antibacterial assays demonstrated that propolis extracts exhibited selective activity, primarily directed against Gram-positive bacteria. Among the five tested strains, clear inhibition zones were observed only for *Staphylococcus aureus* and *Streptococcus mutans*. In contrast, no activity was recorded against Gram-negative bacteria (*Escherichia coli*, *Yersinia enterocolitica*) or the Gram-positive probiotic strain *Lactobacillus plantarum* across the entire concentration range. All images illustrating the inhibition zones are provided in the Appendix A. The negative control (70% ethanol *v*/*v*) showed no inhibitory effect on bacterial growth.

In susceptible strains, the activity of the extracts exhibited a distinct concentration-dependent effect, whereby higher propolis concentrations resulted in larger inhibition zones. Notably, *S. mutans* exhibited markedly higher sensitivity to all active extracts compared with *S. aureus*, as reflected by the significantly larger inhibition zones.

For the susceptible strains, extract activity was clearly concentration-dependent, with higher propolis concentrations producing larger inhibition zones. Both strains were inhibited by the extracts, although the degree of inhibition varied considerably depending on the geographical origin of propolis and the bacterial species. Notably, the raw Uruguayan propolis was completely inactive against *Staphylococcus aureus* at all tested concentrations.

Marked regional differences in antimicrobial activity were observed among the samples. Against *S. aureus* (Table 2), all extracts except the raw Uruguayan sample exhibited significant inhibition. At the highest concentration (200 mg/mL), the Turkish propolis was the most effective, producing a mean inhibition zone of ~12.3 mm. Australian and Romanian samples showed comparable high activity, with inhibition zones of ~10.9 mm and ~10.8 mm, respectively. The Polish extract demonstrated slightly lower, but still notable, potency (~10.2 mm). Overall, the maximum inhibition recorded for all tested extracts was observed against *S. aureus* (12.3 mm for Turkish propolis).

Against *S. mutans* (Table 2), the Turkish sample again displayed the strongest activity (~10.9 mm at 200 mg/mL), followed by the Australian extract (~10.6 mm). Polish (~9.5 mm), Uruguayan ethanolic (~9.3 mm), and Romanian (~9.0 mm) extracts demonstrated lower but measurable activity. Importantly, the ethanolic extract of Uruguayan propolis was not the weakest among the active samples. By contrast, the raw Uruguayan propolis exhibited the lowest activity overall (~8.1 mm at 200 mg/mL) and was completely inactive at concentrations of 100 mg/mL and below inhibition zones.

These findings were confirmed through detailed statistical analysis. Three-way analysis of variance (ANOVA) revealed that the tested factors and their interactions significantly influenced inhibition zone diameters. A highly significant main effect was observed for the geographical origin of propolis (Country; F = 91.27, *p* < 0.001). Importantly, most of the two-way interactions were also statistically significant, including Strain × Country (F = 7.26, *p* < 0.001) and Country × Concentration (F = 2.41, *p* = 0.0099). This underscores the complexity of the phenomenon, in which the effect of a given propolis extract is modulated by the bacterial species and concentration. However, the three-way interaction (Strain × Country × Concentration) was found to be not statistically significant (*p* = 0.103).

Post hoc multiple comparison analysis using Tukey’s HSD test demonstrated that *Staphylococcus aureus* was significantly more sensitive to the tested propolis samples than *Streptococcus mutans*. The mean inhibition zone for *S. aureus* was 10.14 mm, compared with 8.90 mm for *S. mutans*, and this difference (1.24 mm) was highly significant (*p* < 0.001).

Analysis of the Country factor revealed pronounced differences in antibacterial activity. Contrary to the initial assumption, the European samples did not form a homogeneous group. While extracts from Poland (mean inhibition zone: 9.26 mm) and Romania (9.23 mm) showed nearly identical activity (*p* > 0.99), the Turkish extract (10.63 mm) was significantly stronger than both (*p* < 0.001 in each case).

The Turkish extract exhibited the strongest antibacterial activity overall and was significantly more effective than the raw Uruguayan propolis (mean difference = 2.67 mm, *p* < 0.001). The raw Uruguayan sample displayed the weakest effect (mean inhibition zone: 7.96 mm), which was significantly lower than all other tested samples (all *p* < 0.001). The Australian extract (9.49 mm) showed an intermediate activity—significantly stronger than the raw Uruguayan sample (mean difference = 1.53 mm, *p* < 0.001) but significantly weaker than the Turkish extract (mean difference = 1.14 mm, *p* < 0.001).

Comparison of the two Uruguayan preparations further confirmed significant differences. The Uruguayan ethanolic extract demonstrated stronger activity (9.23 mm) than the raw sample from Uruguay (7.96 mm), with a highly significant difference (mean difference = 1.27 mm, *p* < 0.001). These findings clearly confirm that both geographical origin and sample type (raw vs. ethanolic extract) had a strong and statistically significant impact on the antibacterial effect.

Extending this analysis, antibacterial potency was mapped to specific volatiles using Pearson correlation. Against *Streptococcus mutans*, linalool and acetic acid correlated strongly and significantly with larger inhibition zones at 200 and 100 mg/mL (r > 0.90 and r > 0.88), whereas verbenone, α-pinene, trans-pinocarveol, α-cubebene, and camphene correlated very strongly and significantly with weaker effects at 50 mg/mL (r = −0.99 to −0.91). *Staphylococcus aureus* followed the same pattern: linalool was a strong, significant positive predictor at 200 and 100 mg/mL (r ≈ 0.90), while the α-pinene/verbenone cluster was a strong, significant negative predictor (r = −0.98 to −0.90); acetic acid showed a positive but non-significant trend (r ≈ 0.74). Together, these correlations indicate that α-pinene-dominant chemotypes underperform, while linalool-enriched mixtures drive the strongest Gram-positive inhibition, aligning with the superior activity of the ethanolic preparation and the weaker performance of the α-pinene-rich Uruguayan raw sample

## 3. Discussion

In this study, two distinct datasets were subjected to correlation analysis: the volatile profiles of raw propolis obtained by HS-SPME-GC-MS, and the antibacterial activity of its ethanolic extracts. This approach is based on the assumption that the volatile organic compound profile represents a unique “chemical fingerprint” of propolis from a given geographical and botanical origin. Although this fingerprint does not encompass the full range of constituents responsible for bioactivity, it may provide an initial, reliable indicator of the overall chemotype of the sample. This “fingerprint” approach is valuable for initial chemotaxonomic classification and for generating hypotheses about bioactivity, even if the profiled compounds are not the sole bioactive agents. However, this requires confirmation by analyzing a larger number of samples. Nevertheless, the authors acknowledge that direct correlations are inherently limited. We also acknowledge that the use of a single propolis sample from each geographical region represents a primary limitation of this study. While our findings clearly demonstrate distinct chemotypes, further research involving multiple samples per region is necessary to confirm the generalizability of these results and to validate the proposed geographical markers. According to literature [3,30,31,32], the main antibacterial activity of propolis is attributed to non-volatile constituents such as flavonoids (e.g., pinocembrin, galangin) and phenolic acid esters (e.g., CAPE), which are not detectable by GC-MS. Therefore, the correlation analysis presented here is not intended to identify specific volatile compounds as antimicrobial agents but rather to use the volatile profile as a marker of the overall chemotype, allowing prediction and interpretation of the heterogeneous biological activity of ethanolic extracts.

Our results confirm that geographic provenance is a key determinant of both the volatile profile and antibacterial activity. The observed “Europe vs. non-Europe” separation is explained by the dominance of aromatic compounds in poplar-type propolis from the temperate zone and the prevalence of terpenes in non-European samples [7]. Romanian propolis is classified in the literature as poplar-type, consistent with our observations and the benzenoid dominance within the EU cluster [5,7].

Previously published Polish data confirm a phenolic-aromatic poplar-type with high levels of phenolic acid esters and flavanon(e)s, aligning with our European cluster [33]. At the same time, a larger set of Polish 70% EtOH extracts showed a broad range of phenolics (TPC/TFC) and the presence of CAPE, supporting both regional and within-country variability [8].

Two Turkish samples published previously reflect our chemotype division: (i) Ardahan/Erzurum with a poplar-type profile rich in pinocembrin, chrysin, and galangin, consistent with the European aromatic signature, and (ii) Marmaris with a pine-type character dominated by abietane/pimarane diterpenes, which separates it from the EU cluster toward the terpenoid axis; this corroborates our data [34,35].

In the case of Australia, the geographic context is important, with *Eucalyptus* resins and stingless bees contributing predominantly to a terpenoid chemotype, which explains the shift in our Australian samples along the terpene axis in PCA [14].

Microbiologically, our data showed selective activity against Gram-positive bacteria, consistent with multiple reports of higher Gram-positive susceptibility to propolis extracts; a European review (EEP/WEP) confirms lower MICs for Gram-positives and reduced susceptibility of *Pseudomonas aeruginosa*, in agreement with our lack of effect on Gram-negatives [6]. In a Polish series of 15 extracts, the strongest activity was observed against *Bacillus cereus*, whereas *S. aureus* was sometimes non-susceptible in disk diffusion; these findings do not confirm the Nałęczów sample (Poland), where *S. aureus* growth was inhibited, indicating a strong influence of matrix, concentration, and method [8,33]. Additionally, using Pearson correlation in our study, we showed that α-pinene-dominant chemotypes exhibit clearly lower antibacterial activity, whereas linalool-rich mixtures more strongly inhibit Gram-positive bacteria. A complex relationship between the chemical profile and antibacterial activity may be explained by potential synergistic interactions between volatile and non-volatile constituents. It is well documented that lipophilic volatile compounds, particularly monoterpenes such as α-pinene, can disrupt the structural integrity of the bacterial cytoplasmic membrane by penetrating the lipid bilayer [36]. This disruption increases membrane permeability, which in turn may facilitate intracellular access for other antibacterial agents, such as less lipophilic, non-volatile phenolic compounds (e.g., flavonoids, phenolic acid esters) abundantly present in ethanolic extracts [37]. Such a synergistic mechanism could explain why Australian propolis, with its complex terpene-rich profile, exhibited stronger activity compared to the Uruguayan sample, dominated mainly by α-pinene. The broader spectrum of terpenes in the Australian sample may be more effective in permeabilizing the bacterial membrane, thereby enhancing the overall antibacterial efficacy of co-extracted phenolic compounds.

Turkish data from Ardahan/Erzurum demonstrated activity against *S. aureus* and *Bacillus* with no effect on *P. aeruginosa*, matching our Gram-positive selectivity pattern and European literature. It is worth mentioning that MIC values were higher in Erzurum than in Ardahan propolis, correlating with differences in flavonoid levels and supporting our observation that class membership alone does not determine potency and quantitative composition matters [6,34].

Australian reports of strong anti-Gram-positive and antibiofilm effects are concordant with the high efficacy of our Australian samples, supporting a multi-component synergy hypothesis for the terpenoid chemotype [14]. In contrast, the Uruguayan series dominated by α-pinene did not translate into strong bioactivity, suggesting that a high level of a single monoterpene is insufficient without support from a broader mix of terpenes and phenolics; this is consistent with chemotypes defined by multiple markers rather than a single constituent [7].

Methodological differences also contribute to observed discrepancies: we profiled the volatile and semi-volatile fraction by HS-SPME-GC-MS while antibacterial testing was performed on ethanolic extracts; Polish and Turkish studies often used 70% EtOH extracts with phenolic quantification by LC/HPLC, potentially shifting conclusions toward non-volatile phenolics (CAPE, pinobanksin, pinocembrin) and modulating potency—therefore, future work should perform both GC-MS and LC-MS on the same extract used for microbiological assays [8,33,34]. Furthermore, it is plausible that synergistic interactions occur between the volatile terpenes and the non-volatile phenolic compounds found in the ethanolic extracts. For example, certain volatiles could alter bacterial membrane permeability, thereby enhancing the antimicrobial efficacy of phenolic acids and flavonoids [36,37,38].

Our findings support the geography-chemotype-bioactivity model; Bankova’s review links variability in the volatile fraction to local flora and documents broad regional diversity [29]. European poplar-type samples dominated by benzenoids fit the temperate-zone pattern where *Populus* buds are the main source and sesquiterpenes often dominate alongside aromatics. Terpenoid profiles from Australia and Uruguay and Gram-positive selectivity align with evidence that propolis volatile fraction varies in composition and can display antibacterial activity, with effects arising from the interplay of multiple compound classes. The weaker effect of the α-pinene-rich Uruguayan chemotype compared with the more complex Australian profile underscores that a high level of a single monoterpene is insufficient without synergy between volatile and polar constituents.

## 4. Materials and Methods

### 4.1. Chemicals

Ethanol (70%, *v*/*v*), methanol, and acetonitrile were purchased from Sigma-Aldrich (Merck, Darmstadt, Germany). Physiological saline solution (0.9% NaCl), Mueller–Hinton Broth (MH), Mueller–Hinton Agar (MHA), and Brain Heart Infusion (BHI) media were obtained from Oxoid (Thermo Fisher Scientific, Waltham, MA, USA). MRS medium (De Man-Rogosa-Sharpe) used for culturing *Lactobacillus plantarum* was purchased from Merck (Darmstadt, Germany). Sterile paper disks and tetracycline standard disks (30 µg) were supplied by Whatman (Cytiva, UK) and Oxoid, respectively. For chromatographic analyses, helium (99.999%, Air Products, Allentown, PA, USA) was used as carrier gas, along with DVB/CAR/PDMS SPME fibers (50/30 µm, 2 cm long) and an *n*-alkane standard mixture (C7-C40) supplied by Supelco (Merck, Darmstadt, Germany).

### 4.2. Propolis Samples

Three new raw propolis samples were analyzed: Romania (sample ID03, Transylvania), Australia (ID04, stingless bee propolis derived from *Tetragonula carbonaria* and *Tetragonula hockingsi* species from the Queensland regions of northern Australia—Mudgereeba), and one from Uruguay (ID06, Montevideo). Additionally, an ethanolic extract of Uruguayan propolis, registered as a medical product (sample ID05), was obtained and used exclusively for microbiological assays. Previously published data on Polish (ID01) and Turkish (ID02) propolis [9], as well as extract of propolis from Uruguay (ID05, Apiter, Montevideo, Uruguay) were used as comparative reference. Samples were collected in 2023 and stored at 4 °C in glass containers protected from light.

### 4.3. GC-MS Analysis

The volatile fraction of propolis was analyzed using HS-SPME coupled with gas chromatography-mass spectrometry (Agilent 6890 GC/5973 MS, Agilent Technologies, Santa Clara, CA, USA). Approximately 0.5 g of ground propolis sample was placed in 20 mL crimp-top vials sealed with silicone/PTFE septa (Agilent Technologies). Samples were equilibrated at 70 °C for 15 min, followed by adsorption of volatiles on a DVB/CAR/PDMS fiber (50/30 µm, 2 cm; Supelco, Merck, Darmstadt, Germany) for 45 min at 70 °C. Thermal desorption was carried out in the injector at 250 °C. Separation was performed on a ZB-5HT capillary column (60 m × 0.25 mm × 0.25 µm; Phenomenex, Torrance, CA, USA) with helium as the carrier gas at a constant flow rate of 1 mL/min. The oven temperature program was as follows: initial 40 °C for 5 min, increased at 3 °C/min to 250 °C, then at 20 °C/min to 280 °C. The MS was operated in scan mode over the range of 30–450 *m*/*z*, with an ion source temperature of 230 °C and an interface temperature of 300 °C. Compound identification was based on spectral matching with NIST libraries (versions 11 and 14) and linear retention indices (LRI) validation using the Retentify tool [39]. Only compounds with a spectral match of ≥90% were considered. GC-MS relative abundancies (expressed as percentages) were calculated using the chromatogram area normalization method (excluding silane peaks). This standard method presents the content of each compound as a percentage of the total volatile fraction detected and does not represent absolute concentration in the raw sample. Each HS-SPME-GC-MS analysis was performed in triplicate.

### 4.4. Extraction Procedure

Approximately 3 g of each propolis sample (except 04) was ground in a mortar and extracted with 70% ethanol (*v*/*v*) at a 1:8 (*m*/*v*) ratio. Mortars were rinsed with an additional 10 mL of ethanol and transferred to screw-cap glass tubes. In the case of sample 04 (Australia), due to its dense structure, the propolis was first cut into small fragments with a sterile scalpel, then ground in a mortar and subsequently macerated, following the same procedure as for the other samples. Tubes were tightly closed, wrapped in aluminum foil, and incubated under shaking conditions (100 rpm) for 24 h at 50 °C, followed by 24 h at 37 °C. After extraction, samples were precipitated at 4 °C for 24 h. Supernatants were collected and dried using liquid nitrogen (Linde, Poland) in a water bath at 37 °C for 5 h. The obtained dry extracts (0.617–2.017 g) had a resinous, sticky consistency and were stored at 4 °C in the dark.

### 4.5. Preparation of Test Solutions

Stock solutions of dry extracts were prepared in 70% ethanol (*v*/*v*) (Sigma-Aldrich) at a concentration of 200 mg/mL and serially diluted to 100, 50, 25, 12.5, and 6.25 mg/mL. Working solutions were stored in Eppendorf tubes at 4 °C, protected from light.

### 4.6. Bacterial Strains and Media

The following bacterial strains were used: *Escherichia coli* ATCC 8739, *Staphylococcus aureus* ATCC 6538, *Streptococcus mutans* ATCC 25175, and a wild-type *Yersinia enterocolitica* 6471/76-c, a wild-type serotype O:3 strain cured of its virulence plasmid [40]. Additionally, *Lactobacillus plantarum* was included for comparative purposes. Cultures were maintained in Mueller–Hinton Broth (MH), Mueller–Hinton Agar (MHA), and Brain Heart Infusion (BHI) media (Oxoid, Thermo Fisher Scientific, Waltham, MA, USA), while *L. plantarum* was cultured in MRS medium (Merck, Darmstadt, Germany) as presented in Table 3.

### 4.7. Antibacterial Assay

Antibacterial activity was evaluated using the disk diffusion method according to Clinical and Laboratory Standards Institute (CLSI) guidelines. Bacterial suspensions were prepared from 24 h cultures in 0.9% NaCl physiological saline (Oxoid, Thermo Fisher Scientific) and standardized to 0.5 McFarland. Appropriate solid media plates were inoculated with bacterial strains. Sterile paper disks (Whatman, Cytiva, UK) were impregnated with 20 µL propolis extracts, dried under laminar airflow for 20 s, and placed on inoculated agar plates. Negative controls consisted of disks soaked in 70% ethanol (*v*/*v*), while tetracycline standard disks (30 µg, Oxoid) served as positive controls. Plates were incubated at 37 °C for 24 h. Inhibition zones were photographed on millimeter grids and measured with ImageJ 1.54g software (NIH, Bethesda, MD, USA). Each zone was measured three times, and mean values were calculated.

### 4.8. Statistical Analysis

All experiments were performed in triplicate. Results were analyzed using one-way ANOVA, and differences between groups were assessed with Tukey’s HSD test. Statistical analyses were performed with Statistica 13.3 (TIBCO Software Inc., Palo Alto, CA, USA). To evaluate and visualize the chemical similarity among the volatile profiles of the analyzed propolis samples, Hierarchical Cluster Analysis (HCA) was applied. The analysis was conducted on a data matrix comprising five observations (averaged samples) and 50 variables (relative percentage abundances of identified volatile compounds). Ward’s method was used as the clustering algorithm, and Squared Euclidean distances were applied as the dissimilarity measure. The results were presented in the form of a dendrogram. Principal Component Analysis (PCA) was performed to visualize relationships among samples and to identify key compounds driving their differentiation. Results are presented as scatterplots. To further illustrate chemo diversification, a heatmap with bidirectional hierarchical clustering was generated using the MetaboAnalyst 5.0 web platform. This visualization displays the relative abundance of each compound across all samples, with color intensity indicating z-score. To assess the effects of geographical origin, bacterial strain, and extract concentration on antibacterial activity, a three-way analysis of variance (ANOVA) was applied. Group differences were examined with Tukey’s HSD post hoc test. To probe the direct link between chemical composition and bioactivity, Pearson correlation coefficients were calculated between the relative abundance of individual compounds and the mean inhibition zone diameters for *S. aureus* and *S. mutans*.

## 5. Conclusions

The composition of the volatile fraction of propolis is strongly geography-dependent, with two dominant chemotypes: a European type (Poland, Romania, Turkey) enriched in benzenoids, and a non-European type (Australia, Uruguay) rich in terpenes with region-specific signatures. The extracts showed selective activity against Gram-positive bacteria (*S. aureus*, *S. mutans*), with no effect on Gram-negatives or *Lactobacillus plantarum*. The terpenoid-rich Australian propolis displayed the strongest activity, particularly against *S. mutans*, whereas the α-pinene-dominated Uruguayan chemotype was the weakest. The composition-bioactivity relationship was confirmed: benzenoids anchor the European profile, while α-copaene (Australia), α-pinene (Uruguay), and benzoic acid (Europe) can serve as geographic markers for authenticity assessment and activity prediction. These findings support origin-based standardization and direct future work toward coherent profiling of the non-volatile fraction and quantitative MIC/MBC testing, including antibiotic synergy. Crucially, the observed differences in activity warrant follow-up studies involving quantitative MIC/MBC testing to determine the precise potency of these extracts and to explore potential antibiotic synergy. Limitation: volatile profiling by HS-SPME-GC-MS was paired with antibacterial testing of ethanolic extracts, which constrains quantitative correlations; parallel LC-MS of the same microbiologically tested fraction is required. Clinical relevance: terpenoid-enriched extracts high in linalool and low in α-pinene merit evaluation as adjuncts against *S. mutans* within caries-prevention strategies.

## Figures and Tables

**Figure 1 molecules-30-04014-f001:**
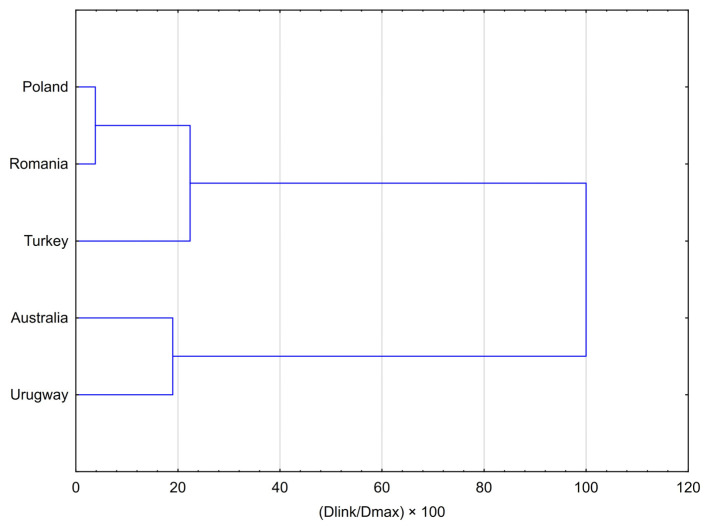
Dendrogram resulting from Hierarchical Cluster Analysis (HCA) for five averaged propolis samples, illustrating the chemical similarity of their volatile profiles. The analysis was performed using Ward’s method as the amalgamation rule and Squared Euclidean distances as the similarity measure.

**Figure 2 molecules-30-04014-f002:**
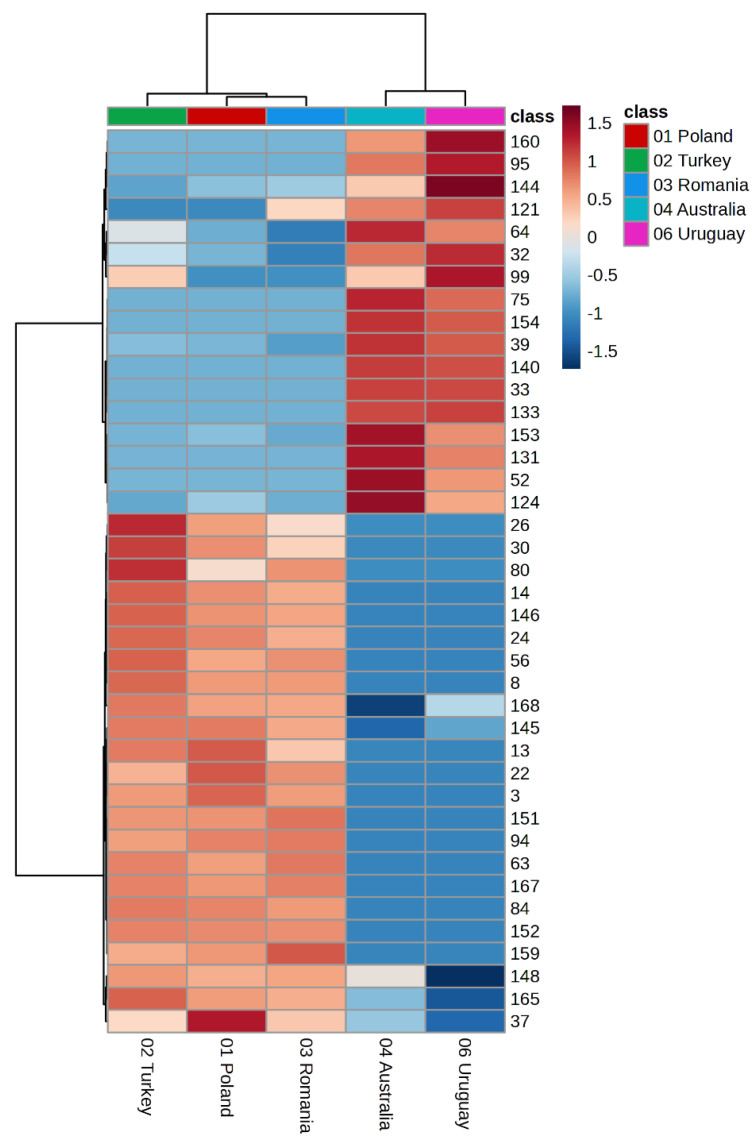
Heatmap with hierarchical clustering (HCA) of volatile compounds in propolis samples. Color intensity reflects the z-value, highlighting regional chemical markers. Variables are numbered according to the numbering of chemicals, as in the GC Table 1. The heatmap with full compound names is provided in the Appendix A.

**Figure 3 molecules-30-04014-f003:**
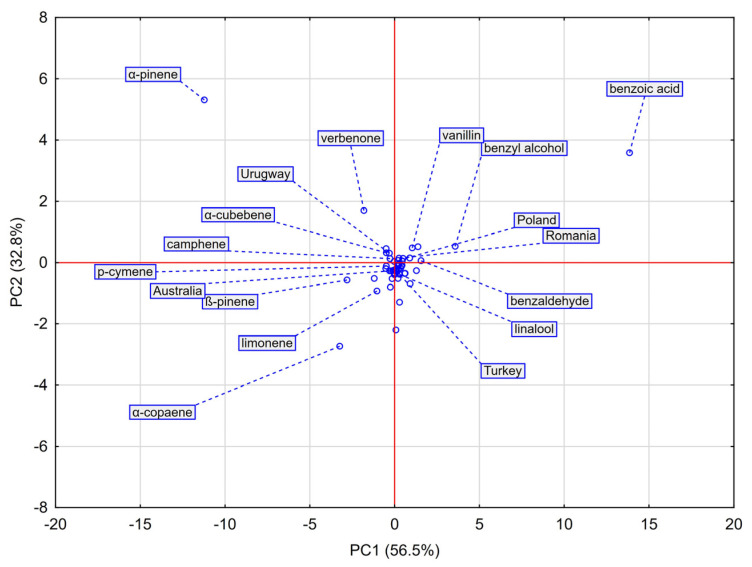
Scatterplot for principal component analysis (PCA) of propolis samples originating from Poland, Romania, Turkey, Australia, and Uruguay based on their chemical composition. PC1 (56.5%) and PC2 (32.8%) together explain 89.3% of the total variance. The distribution of samples highlights associations between geographical origin and the presence of chemical characteristics.

**Table 1 molecules-30-04014-t001:** HS-SPME GC-MS relative abundances (in %) of volatile compounds (area normalization method) from different propolis samples of different origins.

No.	Compound	Linear Retention Index		Sample Origin and Sample Code ID				
				Literature [9]		Original Results		
				Poland	Turkey	Romania	Australia	Uruguay
		LRI _exp_	LRI _ref_	01	02	03	04	06
1	ethanol	nd	427–459	0.48	0.40	0.54	0.73	
2	acetone	nd	475–509	0.31		0.19	0.50	0.37
3	methylene chloride	nd	528–531	1.44	0.49	0.48		
4	hexane	nd	600	5.03	1.77	1.43	2.98	
5	ethyl acetate	609	611	pr	1.22		0.91	0.06
6	acetic acid	612–630	622	1.48	5.39	1.75	4.57	tr
7	ethyl propanoate	691	691					
8	3-methyl-3-buten-1-ol	712	716	0.29	0.77	0.31		
9	methyl isobutyl ketone	723	730					0.01
10	ethyl isobutyrate	748	750					
11	toluene	759	761		0.04		tr	0.38
12	propanoic acid, 2-methyl-	758	758	tr		0.05		
13	2-buten-1-ol 2-methyl-	767	766	0.43	0.26	0.06		
14	2-buten-1-ol, 3-methyl- (=prenol)	769	775	0.48	1.21	0.27		
15	3-methyl-2-butenal	781	783	0.36	0.18	0.27		0.07
16	3-pentanone, 2,4-dimethyl-	794	793				4.91	
17	hexanal	800	800	0.34	0.32	0.37	0.32	
18	ethyl butanoate	802	799					
19	butyl acetate	815	809		tr	0.04		
20	furfural	835	835	0.08	0.08	0.03	0.37	
21	butanoic acid, 2-methyl-, ethyl ester	851	849					
22	trans-2-hexenal	854	853	0.11	0.03	0.05		
23	furfuryl alcohol	858	863				0.08	
24	butanoic acid,2-methyl -(=2-methyl butyric acid)	863	862	0.21	0.31	0.18	tr	
25	*p*-xylene	871	872	0.07	tr		0.17	
26	3-methyl-3-buten-1-ol, acetate	886	883	0.11	0.58	0.03		
27	styrene	893	891	0.21	0.32	0.23	0.94	0.08
28	nonane	900	900	0.09	tr		0.10	
29	heptanal	902	902	0.10	tr	0.06	0.14	
30	2-buten-1-ol, 3-methyl-, acetate (=prenyl acetate)	924	925	0.30	1.26	0.07		
31	α-thujene	929	928	0.06	0.04		0.19	0.09
32	α-pinene	937	936	0.44	1.14	0.18	12.09	28.89
33	α-fenchene	951	949				0.58	0.50
34	camphene	952	950	0.14	0.21		0.30	0.82
35	4-methyl-2-pentenolide *	956	952					
36	thuja-2,4(10)-diene	958	956		0.51		tr	1.86
37	benzaldehyde	965	963	5.93	2.41	2.71	1.36	0.75
38	1-heptanol	972	969				0.18	
39	β-pinene	980	978	0.12	0.13	0.08	7.15	4.54
40	6-methyl-5-hepten-2-one	988	986	0.46	0.11	0.16	0.38	0.10
41	myrcene	992	989	0.38	0.10	0.18	0.13	0.24
42	ethyl hexanoate	999	997		0.43			0.06
43	decane	1003	1000	0.21				
44	octanal	1003	1003	0.27	0.35	0.29	0.34	
45	α-phellandrene	1007	1004		tr	0.08	0.10	
46	3-carene	1013	1011	0.13	0.10			0.05
47	α-terpinene	1020	1017	0.06	0.09	0.03	0.22	0.06
48	*m-*cymene	1025	1022		tr			0.08
49	*p*-cymene	1028	1024	0.29	0.89	0.25	1.27	1.28
50	limonene	1033	1030	0.65	5.32	1.98	3.93	3.62
51	1,8-cineole	1036	1032	0.26	0.22	0.40	0.98	
52	*cis*-β-ocimene	1040	1038				0.67	0.06
53	*o*-cymene	1040	1038					0.16
54	benzyl alcohol	1042	1037	9.39	4.17	5.76	tr	0.30
55	*trans*-β-ocimene	1050	1048	0.21	0.14		0.15	
56	γ-caprolactone (=ethyldihydro-2(3H)-furanone)	1059	1055	0.07	0.21	0.11		
57	trans-2-octenal	1060	1060	0.08	0.06	0.13		0.02
58	γ-terpinene	1063	1060		0.13	0.23	0.32	0.06
59	acetophenone	1072	1067	0.31	0.33	0.24	0.25	
60	*cis*-linalool oxide, (furanoid)	1077	1075	0.40	0.24	0.41	0.34	0.05
61	benzyl formate	1082	1082	0.10	0.20	0.12		0.03
62	*m*-cymenene	1087	1085		0.05			
63	*trans-*linalool oxide, (furanoid)	1092	1083	0.16	0.25	0.30		
64	*p*-cymemene	1094	1089	0.32	0.54	0.23	1.66	1.09
65	heptanoic acid, ethyl ester	1098	1093		0.06			
66	methyl benzoate	1100	1094	0.44	0.16	0.43		0.05
67	linalool	1102	1099	0.64	1.42	0.96	0.81	
68	perillene (=furan, 3-(4-methyl-3-pentenyl)-)	1105	1099					0.59
69	nonanal	1106	1103	0.59	0.16	0.69	0.65	
70	1,3,8-*p*-menthatriene	1117	1112					0.04
71	phenyl ethyl alcohol	1121	1115	3.20	3.30	1.97	tr	0.13
72	methyl octanoate	1125	1128	0.11	0.06	0.01	tr	0.13
73	α-campholenal	1133	1124		0.08			0.78
74	nopinone	1146	1136				0.36	0.12
75	*trans*-pinocarveol	1147	1140		0.28	0.06		1.69
76	camphor (with silane coellution)	1153	1143	2.51	1.06	1.52	1.75	pr
77	(E,E)-2,6-nonadienal	1156	1153	0.04		0.05		
78	unidentified	1158	na	0.10	0.26	0.12		2.69
79	menthone <iso>	1160	1151	0.05	2.35	0.34		
80	trans-2-nonenal	1162	1162	0.08		0.11		
81	benzyl acetate	1168	1166	1.29	1.16	0.54		0.09
82	unidentified	1171	na		1.82	0.37	2.98	0.65
83	ethyl benzoate	1176	1171	0.93	1.11	0.60		
84	menthol	1179	1167		5.95 ^y^	0.46		0.28
85	diethyl succinate	1181	1178		pr			
86	terpinen-4-ol	1185	1177	tr	1.29	0.74	2.30	0.10
87	verbenyl ethyl ether	1188	1186					1.07
88	*p*-cymen-8-ol	1192	1184		1.35	tr	0.14	0.08
89	ethyl octanoate	1197	1196		0.94	0.07	tr	
90	α-terpineol	1198	1190		tr	1.21	2.48 ^x^	tr
91	dodecane	1200	1199		2.55	0.90	tr	0.63
92	methyl salicylate	1202	1193	3.22	1.38	1.02		0.40
93	benzoic acid	1201–1236	1191	25.11	10.55	31.43	pr	
94	myrtenal	1205	1192		tr		0.21	1.26
95	decanal	1207	1205	4.47	tr	1.78	0.13	
96	myrtenol	1212	1194					0.74
97	pin-2-en-4-one *	1220	1204		0.22			0.20
98	Verbenone *	1223	1206		0.11		0.12	5.78
99	carveol, trans-	1227	1217		0.12			1.01
100	β-cyclocitral	1229	1218	0.37	0.38	tr		
101	cumin aldehyde	1251	1238	0.23				0.19
102	carvone	1252	1242		0.19	0.11		0.58
103	geraniol	1257	1255				0.19	
104	2-phenylethyl acetate	1262	1259	0.28	0.59		0.11	
105	geranial	1275	1270			0.03	0.13	
106	*trans*-cinnamaldehyde	1280	1271	0.27		0.59		
107	guaiacol, 4-ethyl-(=phenol, 4-ethyl-2-methoxy-)	1285	1280	0.13			0.06	0.10
108	bornyl acetate	1293	1284		0.23	0.08	0.14	0.42
109	nonanoic ethyl ester	1296	1294					
110	thymol	1296	1290	0.05	1.31	0.14		0.11
111	menthyl acetate	1298	1294		0.39			
112	tridecane	1298	1300	0.04		0.06		
113	carvacrol	1306	1300	0.05	1.92	0.10	0.15	0.32
114	2-propen-1-ol, 3-phenyl- (=*trans*-cinnamyl alcohol)	1315	1312	0.03	0.12	0.46	0.13	0.06
115	*p*-vinyl-guaiacol	1322	1317	0.10	0.04	0.13		0.05
116	myrtenyl acetate	1333	1329		0.03			0.18
117	isobutyl benzoate	1335	1331	0.25	0.05			
118	*trans*-carvyl acetate	1343	1337					0.03
119	hydrocinnamic acid, ethyl ester	1359	1353		0.36			0.06
120	α-cubebene	1359	1351			0.07	0.39	1.37
121	eugenol	1365	1358	0.13	1.21	0.15		0.04
122	α-ylangene	1384	1370	0.06	0.11	0.14	0.37	0.16
123	α-copaene	1388	1376	0.40	0.25	0.27	12.92	2.51
124	decanoic acid, ethyl ester	1397	1391			tr		
125	tetradecane	1399	1400	0.09	0.21	0.20	0.13	
126	β-bourbonene *	1400	1384					1.47
127	β-elemene	1403	1390					1.37
128	vanillin	1411–1414	1405	1.54	0.49	3.00		0.24
129	caryophyllene, cis-	1422	1412	0.03	0.04			0.02
130	longifolene	1424	1415		0.50			
131	α-gurjunene *	1425	1409				0.63	0.09
132	caryophyllene, trans-	1436	1428	0.25	0.31	0.37	1.53	0.49
133	β-copaene	1445	1433				0.30	0.34
134	methyl butyl benzoate	1446	1438	0.57				
135	α-bergamotene	1446	1440		0.21	0.05		
136	α-guaiene	1451	1440	0.58		0.47		0.07
137	cinnamyl acetate	1452	1443		0.05			
138	unidentified	1456	na	1.09		5.53		
139	β-farnesene	1461	1456					0.18
140	cadina-3,5-diene	1466	1458				0.21	0.16
141	α-humulene	1471	1453	0.15	0.13	0.19	0.52	0.10
142	ethyl cinnamate	1476	1465					
143	unidentified	1476	na	0.08		0.25		0.03
144	caryophyllene <9-epi-β-> *	1479	1466	0.14	0.12	0.15	0.24	0.50
145	γ-muurolene *	1490	1476	0.33	0.33	0.31	0.22	0.24
146	ar-curcumene	1491	1482	0.19	0.43	0.14		
147	prenyl benzoate	1494	1485	1.67				
148	β-selinene	1506	1489	0.40	0.59	0.49	0.11	
149	α-muurolene	1513	1509	0.59	0.52		0.34	
150	β-bisabolene	1518	1505		0.16			
151	γ-cadinene	1530	1513	0.43	0.42	1.48		
152	δ-cadinene	1537	1523	0.53	0.60	0.49		
153	*cis*-calamenene	1539	1528	0.54	0.49	0.46	4.11	2.01
154	cadina-1(2),4-diene	1549	1531				0.43	0.22
155	α-cadinene	1553	1533	0.19	0.64			0.18
156	α-calacorene	1555	1544	0.13	0.23	0.14	0.13	
157	β-calacorene	1562	1564		0.03	0.05		0.40
158	*trans*-nerolidol	1572	1564			0.24		0.67
159	hexadecane	1600	1600	0.09	0.06	0.22		
160	spathulenol *	1598	1576				0.10	1.47
161	caryophyllene oxide	1605	1581	0.10	0.05	0.36	0.10	0.39
162	guaiol	1616	1597	0.13	0.11	0.09		0.25
163	1,10-di-epicubenol *	1635	1618	0.06	0.08	0.08		0.15
164	epi-1-cubenol *	1648	1632	0.07	0.12	0.12	0.11	0.27
165	γ-eudesmol	1653	1631	0.79	1.28	0.67	0.12	0.04
166	tau-cadinol (=α-cadinol, epi-)	1660	1648	0.48	0.58			0.49
167	α-muurolol	1665	1668	0.05	0.06	0.06		
168	β-eudesmol	1674	1659	1.16	2.45	1.00	tr	0.06
169	α-eudesmol	1676	1657	0.76	1.08	0.48	tr	0.25
170	cadalene	1695	1671	0.06	0.06	0.06		0.20
171	α-bisabolol	1698	1683		1.01			
172	heptadecane	1701	1700	tr		tr		0.05
173	benzyl benzoate	1784	1766	0.21	0.03	0.42	tr	0.04
174	benzyl salicylate	1847	1867	0.04		0.04		
175	nonadecane	1895	1900			0.02		

The background color was intentionally applied to indicate the compounds showing the highest relative abundances (%) within each propolis sample, as determined by the GC–MS analysis. Legend: nd—not determined, na—not applicable, tr—trace (below 0.01%), pr—present but the % abundance not given due to coellution with another analyte; *—tentative identification, x—coellution with benzoic acid, y—coellution with diethyl succinate, exp—experimental, ref—reference.

**Table 2 molecules-30-04014-t002:** Disk-diffusion inhibition zones (mm) for *Staphylococcus aureus* ATCC 6538 and *Streptococcus mutans* ATCC 25175 produced by 70% ethanol extracts of propolis from Poland, Turkey, Romania, Australia, and Uruguay (extract and raw) at six concentrations (200–6.25 mg/mL). Values are mean ± SD; *n* = 3; 0 indicates no inhibition.

Propolis Origin	Concentration (mg/mL)	*S. aureus*		*S. mutans*	
		Mean	SD	Mean	SD
Poland	200	10.22	0.27	9.47	0.29
	100	10.14	0.30	9.08	0.26
	50	9.56	0.10	8.47	0.02
	25	9.02	0.22	8.09	0.25
	12.5	8.34	0.28	0.00	0.00
	6.25	8.13	0.21	0.00	0.00
Turkey	200	12.34	0.40	10.90	0.54
	100	11.73	0.41	10.12	0.37
	50	10.99	0.37	9.58	0.39
	25	10.63	0.10	8.76	0.64
	12.5	10.09	0.65	8.19	0.23
	6.25	9.18	0.26	8.07	0.15
Romania	200	10.85	0.26	9.01	0.16
	100	10.34	0.10	8.67	0.20
	50	9.41	0.07	8.36	0.16
	25	9.09	0.50	8.14	0.12
	12.5	7.90	0.21	0.00	0.00
	6.25	7.62	0.33	0.00	0.00
Australia	200	10.89	0.98	10.56	0.47
	100	9.79	0.17	9.14	0.36
	50	9.48	0.12	8.50	0.11
	25	9.26	0.07	8.29	0.05
	12.5	9.05	0.09	8.23	0.10
	6.25	8.86	0.28	7.98	0.10
Uruguay standardized extract	200	10.72	0.09	9.32	0.23
	100	10.13	0.24	9.10	0.18
	50	9.36	0.06	8.39	0.09
	25	8.79	0.07	8.02	0.05
	12.5	8.25	0.17	7.94	0.10
	6.25	8.24	0.72	0.00	0.00
Uruguay	200	0.00	0.00	8.12	0.24
	100	0.00	0.00	7.81	0.28
	50	0.00	0.00	0.00	0.00
	25	0.00	0.00	0.00	0.00
	12.5	0.00	0.00	0.00	0.00
	6.25	0.00	0.00	0.00	0.00

Representative photographs of the assay plates are provided in the Appendix A.

**Table 3 molecules-30-04014-t003:** Bacterial strains used in the study and their respective culture media.

Bacterial Strain	Liquid Medium	Solid Medium
*Escherichia coli* ATCC 8739	MH (Mueller–Hinton)	MHA (Mueller–Hinton Agar)
*Staphylococcus aureus* ATCC 6538	MH	MHA
*Streptococcus mutans* ATCC 25175	BHI (Brain Heart Infusion)	MHA
*Yersinia enterocolitica* 6471/76-c (wild-type strain lacking virulence plasmids)	MH	MHA

## Data Availability

The original contributions presented in this study are included in the article/Appendix A. Further inquiries can be directed to the corresponding author(s).

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
