# Peer review of "From Chemistry to Bioactivity: HS-SPME-GC-MS Profiling and Bacterial Growth Inhibition of Three Different Propolis Samples from Romania, Australia, and Uruguay"

_molecules, 2025, doi:10.3390/molecules30194014_

Round 1

Reviewer 1 Report

Comments and Suggestions for Authors

1. In the Materials and Methods section, under "Antibacterial Analysis," the author recommends indicating the propolis extract constant in line 508.
2. When describing strains, the author recommends using the following sources: Escherichia coli ATCC 8739, Staphylococcus 494 aureus ATCC 6538, and Streptococcus mutans ATCC 25175 in the Materials and Methods section of section 4.6. "Strains are lethal and culture media."
3. When describing strains, the author recommends using the following sources: Escherichia coli ATCC 8739, Staphylococcus 494 aureus ATCC 6538, and Streptococcus mutans ATCC 25175 in the Materials and Methods section of section 4.6. In "Strains are harmful to nutrient media," the author notes that in his laboratory he uses the wild-type Yersinia enterocolitica-495 strain, which is virtually devoid of plasmid virulence, to obtain virulence. However, he does not describe the method for obtaining this wild-type strain or provide literature references for its method. The author is advised to describe the method for obtaining the wild-type Yersinia enterocolitica-495 strain in more detail or cite the method from a literary source.
4. When examining Table 1, "HS-SPME GC-MS Comparative Content (%) of Volatile Compounds from Various Propolis Samples of Different Origins," it is not entirely clear what percentage of volatile compounds the author detected in the propolis extracts. The author is advised to indicate the percentage of volatile compounds per 1 g or 1 mg of propolis in the title of this table.
5. The author is advised to note that the numbering in Section 4. Materials and Methods has been inconsistent. After paragraph 4.6. Bacterial strains and conditions after sections 2.7. Antibacterial analysis and 2.8. Statistical analysis, although these sections, it seems to me, belong to section 4. Materials and methods.

Author Response

We would like to sincerely thank Reviewer 1 for the detailed and careful evaluation of the “Materials and Methods” section and for pointing out aspects that required clarification. Your constructive feedback helped us to improve the transparency, consistency, and methodological precision of the manuscript. Below we provide our point-by-point responses.

  1. In the Materials and Methods section, under "Antibacterial Analysis," the author recommends indicating the propolis extract constant in line 508.

We thank the reviewer for this comment. We clarified that 20 µL of extract was used for disk impregnation, ensuring complete saturation of a standard 6 mm disk without oversaturation. This correction has been introduced in Section 4.7, line 535.

  1. When describing strains, the author recommends using the following sources: Escherichia coli ATCC 8739, Staphylococcus 494 aureus ATCC 6538, and Streptococcus mutans ATCC 25175 in the Materials and Methods section of section 4.6. "Strains are lethal and culture media."
  2. When describing strains, the author recommends using the following sources: Escherichia coli ATCC 8739, Staphylococcus 494 aureus ATCC 6538, and Streptococcus mutans ATCC 25175 in the Materials and Methods section of section 4.6. In "Strains are harmful to nutrient media," the author notes that in his laboratory he uses the wild-type Yersinia enterocolitica-495 strain, which is virtually devoid of plasmid virulence, to obtain virulence. However, he does not describe the method for obtaining this wild-type strain or provide literature references for its method. The author is advised to describe the method for obtaining the wild-type Yersinia enterocolitica-495 strain in more detail or cite the method from a literary source.

We thank the reviewer for these important remarks and would like to address them together. We agree that the description of Yersinia enterocolitica was insufficient. We have revised the text to provide the exact strain designation and included appropriate literature references concerning its origin and characteristics. Furthermore, information on the other bacterial strains, including the ATCC codes (E. coli ATCC 8739, S. aureus ATCC 6538, S. mutans ATCC 25175), is now explicitly stated in Section 4.6. We believe that the strain information is now precise and complete.

  1. When examining Table 1, "HS-SPME GC-MS Comparative Content (%) of Volatile Compounds from Various Propolis Samples of Different Origins," it is not entirely clear what percentage of volatile compounds the author detected in the propolis extracts. The author is advised to indicate the percentage of volatile compounds per 1 g or 1 mg of propolis in the title of this table.

Thank you for this valuable remark. We would like to clarify that the percentages presented in Table 1 refer to the relative abundances of volatile compounds, calculated by the peak area normalization method (excluding silane peaks), as described in the Materials and Methods section. These values do not represent the absolute content of volatiles per gram or milligram of propolis, but rather the proportional contribution of each compound to the overall volatile profile, which is the standard approach for HS-SPME-GC analyses.

To avoid any misunderstanding, we have modified the table title to:

“HS-SPME GC-MS relative abundances (in %) of volatile compounds (area normalization method) from different propolis samples of different origins.”

Additionally, we inserted a clarifying sentence in Section 4.3, lines 503–504: “This standard method presents the content of each compound as a percentage of the total volatile fraction detected and does not represent absolute concentration in the raw sample.”

  1. The author is advised to note that the numbering in Section 4. Materials and Methods has been inconsistent. After paragraph 4.6. Bacterial strains and conditions after sections 2.7. Antibacterial analysis and 2.8. Statistical analysis, although these sections, it seems to me, belong to section 4. Materials and methods.

We sincerely thank the reviewer for spotting this formatting error. We have corrected the numbering of subsections, and the sections for “Antibacterial assay” and “Statistical analysis” are now correctly numbered as 4.7 and 4.8, respectively.

We are grateful for your constructive comments, which allowed us to clarify methodological details, correct inconsistencies, and improve the overall transparency of the manuscript. All changes are clearly indicated in the revised version using track changes.

Reviewer 2 Report

Comments and Suggestions for Authors

This manuscript presents a well-structured and comprehensive comparative study on the volatile profiling and antibacterial activity of propolis from three distinct geographical origins. The research is timely, methodologically sound, and addresses a relevant gap in the field of propolis research. The use of HS-SPME-GC-MS for volatile analysis and disk diffusion assays for antibacterial evaluation is appropriate. The chemometric analyses (HCA, PCA) are rigorously applied and support the geographical clustering of samples. The findings contribute meaningfully to the understanding of geography-chemotype-bioactivity relationships. Below comments are provided for its improvement.

  1. The study relies on single samples, which limits the generalizability of the findings. Including multiple samples per region would strengthen the conclusions regarding geographical markers.
  2. The antibacterial assays were performed on ethanolic extracts, while volatile profiling was done on raw propolis. Although justified, this disjunction limits direct correlation between chemical composition and bioactivity.
  3. The antibacterial assays were performed in triplicate, which is acceptable, but increasing the number of biological replicates would improve robustness.
  4. The authors rightly note that non-volatile compounds are major contributors to antibacterial activity. A brief discussion on how these might interact with volatiles would enrich the narrative.
  5. The heatmap (Figure 2) is informative but could benefit from clearer labeling of compound names or IDs for better interpretability.
  6. In Table 1, some compounds are marked without quantitative values. Whenever possible, approximate values or ranges should be provided.
  7. Why not additionally choose MIC and MBC assay?

Author Response

REV2

We sincerely thank Reviewer 2 for the positive and constructive evaluation of our manuscript. Your thoughtful feedback helped us to further refine the discussion, highlight the study’s limitations, and strengthen the overall scientific narrative. Below we provide detailed point-by-point responses.

This manuscript presents a well-structured and comprehensive comparative study on the volatile profiling and antibacterial activity of propolis from three distinct geographical origins. The research is timely, methodologically sound, and addresses a relevant gap in the field of propolis research. The use of HS-SPME-GC-MS for volatile analysis and disk diffusion assays for antibacterial evaluation is appropriate. The chemometric analyses (HCA, PCA) are rigorously applied and support the geographical clustering of samples. The findings contribute meaningfully to the understanding of geography-chemotype-bioactivity relationships. Below comments are provided for its improvement.

  1. The study relies on single samples, which limits the generalizability of the findings. Including multiple samples per region would strengthen the conclusions regarding geographical markers.

We fully agree with the reviewer. This study was designed as a preliminary comparative analysis. To clearly emphasize this limitation, we have added a statement in the Discussion noting that further investigations with a larger number of samples are needed to confirm the generalizability of our conclusions.

  1. The antibacterial assays were performed on ethanolic extracts, while volatile profiling was done on raw propolis. Although justified, this disjunction limits direct correlation between chemical composition and bioactivity.

We thank the reviewer for highlighting this important methodological aspect. We have addressed this limitation in the Discussion, clarifying our rationale for using the volatile profile as a “chemical fingerprint” for preliminary chemotype classification. We also reinforced this paragraph to better present the value of this approach in generating hypotheses regarding bioactivity.

  1. The antibacterial assays were performed in triplicate, which is acceptable, but increasing the number of biological replicates would improve robustness.

We agree that additional biological replicates always improve the reliability of the results. Our assays were performed with three technical replicates, which is the standard and accepted practice for the disk diffusion method. We acknowledge, however, that future validation studies would benefit from including a larger number of biological replicates.

  1. The authors rightly note that non-volatile compounds are major contributors to antibacterial activity. A brief discussion on how these might interact with volatiles would enrich the narrative.

We enriched the Discussion by adding a short hypothesis on the potential synergistic interactions between volatile terpenes and non-volatile phenolic compounds, for example through mechanisms such as increased bacterial membrane permeability

  1. The heatmap (Figure 2) is informative but could benefit from clearer labeling of compound names or IDs for better interpretability.

We agree with this suggestion. To improve readability without overloading the figure, we revised the legend of Figure 2 to more clearly direct the reader to Table 1, where the numeric IDs from the heatmap correspond to full compound names. Additionally, we prepared an alternative version of the heatmap with full compound names, which has been included in the Supplementary Materials (Figure S1). This solution was chosen because, at the typical publication size, the figure becomes visually overcrowded, and the MetaboAnalyst software has limitations regarding label length.

  1. In Table 1, some compounds are marked without quantitative values. Whenever possible, approximate values or ranges should be provided.

We have updated the legend of Table 1, which now contains precise definitions: “tr – trace (relative content <0.01%), pr – present, but % content not provided due to co-elution with another analyte.”

  1. Why not additionally choose MIC and MBC assay?

We selected the disk diffusion method as a robust and well-established tool for preliminary screening and comparative purposes, which was the main aim of this study. We fully agree that MIC and MBC assays represent a logical next step. We strengthened this point in the Conclusions, stating that our results provide strong justification for future quantitative studies employing MIC and MBC determinations.

In line with the reviewer’s valuable suggestions, our future research will therefore focus on expanding the number of propolis samples from each geographical region to strengthen the generalizability of the conclusions. Moreover, integrating quantitative antibacterial assays such as MIC and MBC, as well as exploring the interplay between volatile and non-volatile constituents, will provide deeper mechanistic insight into the synergistic drivers of antibacterial activity.

We are grateful for the reviewer’s insightful comments, which allowed us to better articulate the study’s limitations, expand the discussion, and improve figure and table clarity. All suggested revisions have been implemented in the manuscript and are highlighted using track changes.

Reviewer 3 Report

Comments and Suggestions for Authors

Title: From Chemistry to Bioactivity: HS-SPME-GC-MS Profiling and Bacterial Growth Inhibition of Three Different Propolis Samples from Romania, Australia, and Uruguay

This study presents a comprehensive comparative analysis of the volatile profiles and antibacterial properties of propolis from Romania, Australia, and Uruguay, benchmarked against previously published data from samples from Poland and Turkey. A total of 175 volatile compounds were identified and relatively quantified in volatile profiles of five propolis samples collected from different geographical regions using HS-SPME-GC-MS. Dendrogram resulting from Hierarchical Cluster Analysis (HCA) for five averaged propolis samples, illustrating the chemical similarity of their volatile profiles were also presented. The authors also presented Scatterplot for principal component analysis (PCA) of propolis samples highlighting associations between geographical origin and the presence of chemicals characteristics. The authors also tested the antibacterial activity using disc diffusion method. The authors were also aware of the limitations of the study based on the analytical instruments used and the antibacterial method chosen and it was excellently addressed. The data on geographical variation based on samples from different continents are very interesting. The paper is of high quality and therefore publishable in this quality journal in its current form. Congratulations to the authors.

Author Response

REV3

We are deeply grateful to Reviewer #3 for the very positive and highly encouraging evaluation of our manuscript. Your kind words and recommendation for publication are extremely valuable to us.

This study presents a comprehensive comparative analysis of the volatile profiles and antibacterial properties of propolis from Romania, Australia, and Uruguay, benchmarked against previously published data from samples from Poland and Turkey. A total of 175 volatile compounds were identified and relatively quantified in volatile profiles of five propolis samples collected from different geographical regions using HS-SPME-GC-MS. Dendrogram resulting from Hierarchical Cluster Analysis (HCA) for five averaged propolis samples, illustrating the chemical similarity of their volatile profiles were also presented. The authors also presented Scatterplot for principal component analysis (PCA) of propolis samples highlighting associations between geographical origin and the presence of chemicals characteristics. The authors also tested the antibacterial activity using disc diffusion method. The authors were also aware of the limitations of the study based on the analytical instruments used and the antibacterial method chosen and it was excellently addressed. The data on geographical variation based on samples from different continents are very interesting. The paper is of high quality and therefore publishable in this quality journal in its current form. Congratulations to the authors.

We are pleased that you considered our study to be well-structured, comprehensive, and of high quality. We also appreciate that you recognized the strengths of our methodology, the clarity of our chemometric analyses, and our discussion of study limitations. Your encouraging feedback and recommendation for publication are a great motivation for our further research.

We sincerely thank you once again for your supportive review and recommendation for acceptance. We are grateful for the time and effort you devoted to the evaluation of our work.

Round 2

Reviewer 2 Report

Comments and Suggestions for Authors

The MS is suitable for publication